# Pharmacometric Evaluation of Umbilical Cord Blood Concentration-Based Early Initiation of Treatment in Methadone-Exposed Preterm Neonates

**DOI:** 10.3390/children8030174

**Published:** 2021-02-25

**Authors:** Samira Samiee-Zafarghandy, Tamara van Donge, Karel Allegaert, John van den Anker

**Affiliations:** 1Division of Neonatology, Department of Paediatrics, McMaster University, Hamilton, ON L8S 4L8, Canada; 2Division of Pediatric Pharmacology and Pharmacometrics, University Children’s Hospital Basel (UKBB), University of Basel, 4056 Basel, Switzerland; tamara.vandonge@ukbb.ch (T.v.D.); jvandena@childrensnational.org (J.v.d.A.); 3Department of Clinical Pharmacy, Erasmus MC, Postbus 2040, 3000 CA Rotterdam, The Netherlands; karel.allegaert@uzleuven.be; 4Department of Development and Regeneration, KU Leuven, Herestraat 49, 3000 Leuven, Belgium; 5Department of Pharmaceutical and Pharmacological Sciences, KU Leuven, Herestraat 49, 3000 Leuven, Belgium; 6Division of Clinical Pharmacology, Children’s National Health Hospital, Washington, DC 20010, USA; 7Intensive Care and Department of Pediatric Surgery, Erasmus MC Sophia Children’s Hospital, 3015 GD Rotterdam, The Netherlands

**Keywords:** preterm neonate, methadone, neonatal abstinence syndrome, dosing

## Abstract

In methadone-exposed preterm neonates, early identification of those at risk of severe neonatal abstinence syndrome (NAS) and use of a methadone dosing regimen that can provide effective and safe drug exposure are two important aspects of optimal care. To this end, we reviewed 17 methadone dosing recommendations in the international guidelines and literature and explored their variability in key dosing strategies. We selected three of the reviewed dosing regimens for their pharmacokinetics (PK) characteristics and their exposure–response relationship in three gestational age groups of preterm neonates (28, 32 and 36 gestational age weeks) at risk for development of severe NAS (defined as an umbilical cord methadone concentration of ≤60 ng/mL, following fetal exposure). We applied early (12 h after birth) vs. typical (36 h after birth) initiation of treatment. We observed that use of universally recommended dosing regimens in preterm neonates can result in under- or over-exposure. Use of a PK-guided dosing regimen resulted in effective target exposures within 24 h after birth with early initiation of treatment (12 h after birth). Future prospective studies should explore the incorporation of umbilical cord methadone concentrations for early identification of preterm neonates at risk of developing severe NAS and investigate the use of a PK-guided methadone dosing regimen, so that treatment failure, prolonged length of stay and opioid over-exposure can be avoided.

## 1. Introduction

The increasing incidence of opioid use disorder (OUD), reaching epidemic proportions in the United States, has been associated with a substantial increase in the national prevalence of opioid misuse among women of reproductive age [1,2]. Methadone has been recommended since the early 1990s for improvement of maternal and neonatal outcomes in pregnant women with OUD [3,4,5]. In utero exposure to methadone is associated with an increased risk of preterm birth and can result in neonatal abstinence syndrome (NAS), with up to 40% of methadone-exposed preterm neonates developing NAS [6,7].

The increasing number of these most vulnerable victims of NAS, along with the incremental health care burden with protracted length of stay and mounting health care costs, has resulted in large efforts in order to optimize the care model of this highly challenging disease [8].

To date, the majority of the available studies, however, focused on narcotic-exposed term neonates with limited data available on optimal diagnosis and management of NAS in preterm neonates.

Although methadone is one of the most common opioids, used as first-line treatment for NAS [9], data on its pharmacokinetic/pharmacodynamic (PK/PD) relationship in preterm neonates, are scarce. This results in in a multitude of dosing regimens and variability in clinical practice [10]. Lack of sufficient information to guide precision dosing for methadone in preterm neonates with NAS is a significant barrier in the optimal management of this condition [11]. 

Historically, the decision for the pharmacologic treatment of opioid-exposed neonates, has been dependent on the severity of withdrawal symptoms assessed using the Finnegan Neonatal Abstinence Scoring (FNAS) Tool [12]. To date, use of this tool to identify opioid-exposed preterm neonates in need for pharmacologic treatment has not been validated. It has been suggested that FNAS-driven interventions can result in delayed initiation of the pharmacological treatment, inefficient dosing modification, and slowdown of methodological weaning in preterm neonates, whose hunger-related cues maybe suppressed [12,13]. All of which can result in delayed initiation of treatment due to poorer recognition, a high cumulative dose, prolonged opioid exposure and protracted length of stay [14].

There is evidence that umbilical cord or postnatal methadone concentrations might be correlated with the development and severity of NAS [15,16,17]. In 1976, Rosen and Pippenger studied the relationship between maternal and neonatal plasma concentrations of methadone and showed that neonates with plasma methadone concentrations of ≥60 ng/mL on day zero of life, appeared to be protected from withdrawal [15]. Years later, Kuschel et al. showed that infants who required pharmacologic treatment for NAS had significantly lower umbilical cord methadone concentrations than infants who did not require treatment (31 vs. 88 ng/mL, *p* = 0.029). Furthermore, none of the infants with postnatal plasma methadone concentrations ≥ 20 ng/mL at 48 h of age required pharmacologic treatment for their withdrawal symptoms [16]. In a pilot study on PK of oral methadone in term neonates for the treatment of NAS, median area under the concentration–time curves (AUCs) at 24 and 48 h of 816 and 2274 ng·h/mL, respectively, were shown to correlate with normalization of FNAS scores (scores < 8) [18]. These findings indicate the importance of incorporation of the PK/PD relationship of methadone into strategies for optimization of pharmacologic care of neonates with NAS.

In the current study, we aimed to (i) investigate the variability in the recommended dosing regimens of methadone in the international guidelines and the available literature, (ii) assess the PK parameters of selected recommended dosing regimens in three gestational age (GA) groups of preterm neonates at risk for developing severe NAS and (iii) explore how the PK parameters of these commonly practiced dosing regimens relate to the available exposure-response data [18].

For the purpose of the current simulation study, we defined preterm neonates at risk for severe NAS as neonates ≤36^+6^ weeks GA with assumed umbilical cord methadone concentrations of <60 ng/mL [16].

## 2. Materials and Methods

### 2.1. Assessment of Variability in Dosing Regimens

We selected seven neonatal methadone dosing regimens from international neonatal drug formularies and 10 dosing regimens reported in the literature [19,20,21,22,23,24,25,26,27,28,29,30,31,32,33]. We recorded the following variables for each dosing regimen: utilization of different treatment phases (initial phase, escalation phase, maintenance phase, weaning phase); dose per administration, dosing interval and total daily dose in each treatment phase; and factors used for the selection of a priori dosing regimen, i.e., demographic characteristics or use of NAS score.

### 2.2. Simulation of Methadone Exposure in Three Selected Dosing Regimens

To date, limited population PK models for methadone in preterm neonates have been published [10,26]. We applied the population PK model of van Donge et al. as the population investigated in this study was exclusively preterm neonates and methadone was administered shortly after birth (median postnatal age of 3 days) [10]. PK data of this single-center open-label prospective study included 29 preterm neonates which were exposed to a single dose of orally administered methadone. One dose of standard opioid medication (fentanyl or morphine) which was prescribed for clinical reasons was replaced by one dose of 0.1 mg/kg methadone. The median gestational age and body weight amounted 32 weeks (range 26 to 36) and 1.6 kg (range 0.93 to 2.7), respectively. The data were best described by a one-compartment model with linear elimination. Clearance and volume of distribution were both influenced by GA. Apparent volume of distribution and clearance for (R)-methadone and (S)-methadone was estimated to be 26.9 L and 0.244 L/h, and 18 L and 0.167 L/h, respectively. For further details on the population PK model, we refer to the initial study [10]. To generate individual methadone concentration-time profiles, the demographic characteristics of three typical preterm neonates (28, 32, 36 weeks of GA and 1.25, 1.6, 2.3 kg birth weight, respectively) were used. The demographic characteristics of GA and body weight of the patients in the original study (median of 32 weeks and 1.6 kg) were selected as we aimed to use their corresponding PK characteristics in our current model. Each individual was simulated 1000 times including between-subject variability. We assumed an in utero exposure to methadone only with a neonatal methadone concentration approximating 30 ng/mL at birth to target the population that have the highest risk for the development of NAS and are in need for pharmacological intervention (< 60 ng/mL) [15,16,17]. We selected two dosing regimens from the international guidelines: Neofax (minimum [0.05 mg/kg q24 h] and maximum [0.2 mg/kg q12 hr] recommended dosing regimen) and one dosing recommendation from the literature (van Donge et al. [Day 1: 0.1 mg/kg q6 h, Day 2: 0.1 mg/kg q12 h, Day 3: 0.05 mg/kg q12 h, Day 4–7: 0.01 mg/kg q24 h]) for the assessment of their PK characteristics and the relationship of their PK parameters to the available PK/PD data [10,15,16,17]. Treatment initiation time was set to 12 h and 36 h after birth and treatment duration to 72 h, to simulate early treatment initiation (12 h) versus standard practice (36 h). The early treatment initiation was chosen at 12 h after birth to ensure feasibility of access to umbilical cord methadone concentration as a potential predictor. Simulations were performed using nonlinear mixed effects modelling software (NONMEM v7.4.1; ICON Development Solutions, Ellicott City, MD, USA) [34]. We investigated the plasma concentration and AUC within the 72 h after birth, as surrogate of response for each of the assessed dosing regimens at the two selected treatment initiation times.

## 3. Results

### 3.1. Variability in Neonatal Methadone Dosing Guidelines

Only one of 17 studied dosing regimens provided specific data to preterm neonates. We observed substantial variability in the 17 dosing regimens studied (seven from international guidelines and ten provided from the literature), with respect to the starting dose, dosing interval, total daily dose, distinct phases of treatment (initial, escalation, maintenance and/or weaning phase) and use of NAS score as a priori dosing regimen (14 = Yes vs. 3 = No) (Table 1). Of the international guidelines, only the dosing regimens from Neofax did not use stratification based on the NAS scoring system. The starting dose differed substantially and ranged from 0.05 mg/kg to 0.8 mg/kg between regimens, with the most common starting dose of 0.1 mg/kg (9, 53%) [1,10]. The dosing interval strongly varied from every 4 h to every 24 h. The total daily dose for the initial phase ranged from 0.05 to 2.4 mg/kg. Treatment phases were not specified in all dosing guidelines.

### 3.2. Methadone Exposure

Applying the dosing regimen of van Donge et al. to typical preterm neonates of three GA groups, at risk for development of NAS in need for pharmacologic treatment (presumed umbilical cord methadone concentration of approximating 30 ng/mL) resulted in a cumulative AUC at 24 and 48 h after birth comparable with the target AUC for control of withdrawal symptoms in all the three GA groups, but only if the methadone therapy was initiated at 12 h of age (target exposure: a median (AUC) at 24 and 48 h of 816 and 2274 ng·h/mL, respectively) [18]. Treatment initiation at 36 h of age resulted in decrease in plasma methadone concentration, reaching a nadir at 24 h of life, and failed to reach an adequate exposure until 72 h after birth (Table 2).

Application of a Neofax minimum recommended dosing regimen demonstrated that this dosing regimen does not reach the target exposure in either of the three groups of preterm neonates within the first 48 h after birth, even with treatment initiation as early as 12 h of age (Table 3). The Neofax maximum methadone dosing regimen reached the target exposure at 24 h and onward, in all three gestational age groups. We observed a 20% higher methadone exposure as compared to the dosing regimen of van Donge et al., with a median cumulative AUC of 3786 vs. 3141 ng·h/mL for Neofax maximum dosing regimen and van Donge et al., in a typical neonate of 36 weeks GA, respectively [10,22].

The overall methadone exposure was higher for the most immature neonate, explained by GA-dependent clearance capacity, with lower methadone clearance in the most immature preterm infants [10]. Median predicted individual methadone concentration-time profiles, with treatment initiation at 12 h of life during the first 72 h of life, showed postnatal plasma concentration of above 20 ng/mL at 48 h of life, for all the three applied dosing regimens, in all the three groups of preterm neonates (Figure 1).

## 4. Discussion

In the current study, we demonstrated high variability in key dosing strategies in the recommended dosing regimens of methadone for NAS disease. Furthermore, we depicted how the available dosing recommendations could create under-exposure or over-exposure to methadone in preterm neonates of ≤36 weeks GA. As Neofax dosing regimen was one of the three guidelines that did not use NAS scoring based dosing stratification, it was feasible for us to perform the PK simulation study without access to the NAS scoring data [22]. The dosing regimen by Sui et al., also did not implement the usage of NAS scoring and recommended the exact dosing schedule as Neofax [31]. For dosing recommendations which were based on the NAS scoring, we were unable to investigate the exact degree of exposure in preterm neonates. We, however, observed equal or higher initial dosing with advice for further escalation, as compared to Neofax maximum regimen, in 8 out of the remaining 14 recommendations [19,21,23,25,27,28,32,33]. As per our data, use of any of these dosing regimens, can result in substantially high exposure to methadone in preterm neonates of ≤36 weeks GA. In view of the current scarcity of data on short- and long-term adverse events of methadone in preterm neonates, use of such universal dosing recommendations in methadone-exposed preterm neonates can be a serious concern.

Methadone-associated NAS often requires prolonged hospitalization and high postnatal cumulative opioid treatment. Evidence has shown early identification of neonates in need for pharmacological treatment and optimal dosing strategy may decrease the severity of NAS and length of stay [10,35,36,37]. However, when based on clinical tools like the FNAS, recognition of NAS in preterm neonates is substandard [13].

We observed the dosing recommendation of van Donge et al., which was the only dosing regimen designed with integration of developmental PK data in preterm neonates, provided cumulative exposures comparable to the suggested target exposures, in term neonate, for optimal PK/PD relationship [10,18]. Our result emphasizes the importance of incorporation of the PK/PD relationship data in designing the optimal dosing regimens of methadone, to avoid treatment failure and subsequently prolonged length of hospital stay or unnecessarily high exposure and occurrence of adverse events [11].

Limited data in opioid-exposed term neonates, have suggested the correlation of umbilical cord methadone concentration with incidence and severity of NAS in need for pharmacologic treatment [15,16,17]. In our simulation study, we assumed a baseline methadone level of <60 ng/mL, to target the population at highest risk for development of NAS requiring pharmacologic therapy. We observed that early initiation of treatment (12 h after birth) can provide target therapeutic exposures as early as 24 h after birth. On the contrary, when we applied a time for treatment initiation that was more reflective of the common practice (36 h post birth) [18], the target therapeutic exposure was only achieved at 72 h post birth. This result suggests that early initiation of treatment might prevent the abrupt cessation of chronic opioid exposure and maintains a methadone plasma concentration that, as per the available studies, correlates with the effective prevention or control of the symptoms of NAS within the first 24 h of life [18].

Such early initiation of treatment requires a validated tool for an early identification of patients at high risk for NAS disease in need for pharmacologic treatment. The utility of umbilical cord methadone concentration, in the development of such a tool, is a highly interesting and important area in need for prospective investigations in both preterm and term methadone-exposed neonates. The inclusion of various scoring systems in these suggested studies are essential to relate the exposure of methadone to the clinical response. Such a clinical scoring system should be developed and validated, considering symptoms that are specific to preterm neonates for opioid withdrawal, while addressing other preterm medical diagnoses [36]. Evaluating the performance of this tool, per se and as an adjunctive tool, for risk prediction of NAS with highest sensitivity, specificity, negative and positive predictive value in a prospective study is of utmost importance.

Although preterm neonates are historically believed to have a lower risk for the development of NAS, previous studies have shown that up to 40% of opioid-exposed preterm neonates develop NAS, with a maximum score comparable to term neonates (10 vs. 11) [36,38]. Lack of a standardized scoring tool can at least partly contribute to under-recognition of incidence and severity of NAS in preterm neonates, which along with use of universal methadone dosing regimens, without integration of developmental PK, results in suboptimal care of these patients [36].

The use of data on umbilical cord methadone concentrations, postnatal plasma concentrations and target exposures from limited available evidence, in terms of methadone-exposed neonates, is a limitation. Although, such data do not exist in preterm neonates, available evidence on cord plasma concentrations of methadone-exposed term and preterm neonates did not show any significant difference [17]. Furthermore, in view of a lack of data specific to preterm neonates, extrapolation of data is unavoidable to expand the evidence required to optimize the care of such vulnerable populations [39]. Our assumption for initiation of pharmacologic treatment in preterm neonates with umbilical cord methadone concentration of less than 60 ng/mL has to be prospectively studied and validated. Future studies might consider complementing this strategy with the results of standardized scoring, so that the sensitivity and specificity of such a screening tool can be further optimized.

## 5. Conclusions

Methadone-exposed preterm neonates can be at risk of under- or over-treatment with the use of the currently recommended methadone dosing regimen. Integration of developmental PKs in designing a dosing regimen specific to preterm neonates, along with early initiation of treatment can help overcome this issue. Umbilical cord methadone concentrations appear as a promising tool to assist early identification of neonates at risk for developing NAS in need for pharmacologic treatment. To best optimize the care model for the most vulnerable victims of this challenging disease, development of a highly sensitive and specific screening tool along with the best incorporation of PK/PD relationship should be investigated in a prospectively designed study.

## Figures and Tables

**Figure 1 children-08-00174-f001:**
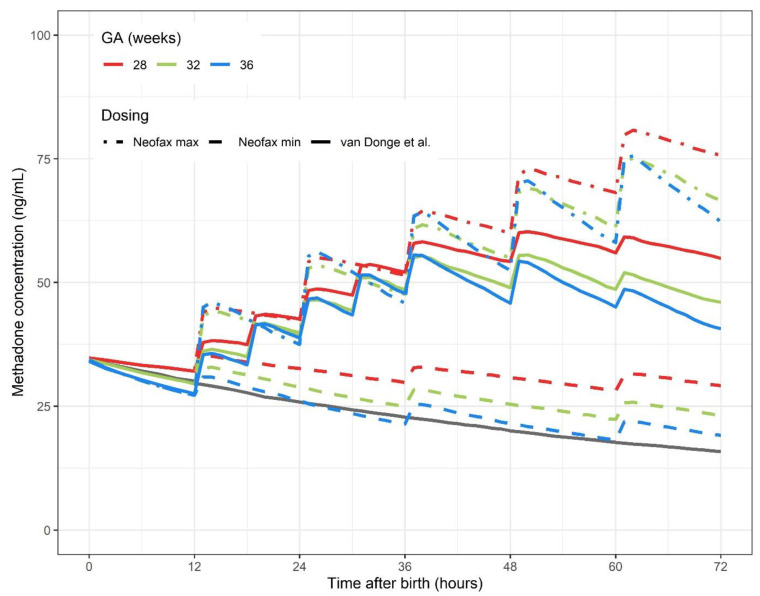
Median predicted methadone concentration (ng/mL) for three typical preterm neonates of 28, 32 and 36 weeks, applying three methadone dosing regimens. Median predicted methadone concentration-time profile for a typical preterm neonate of 32 weeks not receiving methadone treatment is shown in the straight grey line. GA gestational age.

**Table 1 children-08-00174-t001:** Methadone dosing regimens for the treatment of neonatal abstinence syndrome collected from international guidelines and literature.

	Initial Phase *	Escalation Phase	Maintenance/Stabilization Phase	Weaning Phase	NAS Score	Total Daily Dose ^$^
**Dosing Recommendations International Guidelines**						
BNFc 2011–2012 [19]	0.1 q6 h	↑ by 0.05 q6 h until symptoms are under control	Total daily dose that controls symptoms divided over 2 doses	Reduce over 7–10 days	Yes	0.4
Nelson Textbook of Pediatrics 2015 [20]	0.1 q12 h	↑ by 0.025 q4 h (score > 8) maximum of 0.5	Total daily dose that controls symptoms divided over 2 doses	Decrease dose by 10% every 1–2 weeks, discontinue when dose is 0.05	Yes	0.2
Frank Shann’s Drug Doses 2017 [21]	0.1–0.2 q6 h/q12 h	-	-	-	Yes	0.2–0.8
Neofax 2010 [22]	0.05–0.2 q12 h/q24 h	-	-	10–20% decrease every week, over 4–6 weeks	No	0.05–0.4
Neonatal formulary 7 2011 [23]	0.1 q6 h	↑ by 0.05 q6 h until symptoms are under control	Total daily dose that controls symptoms divided over 2 doses (q12 h)	Sustained control for 48 h -> reduce dose by 10–20% every day	Yes	0.4
VCHIP Neonatal Guideline [24]	0.3–0.6 mg q12 h	↑ by 0.05–0.2 mg (scores ≥ 9 after 4 doses)	-	If a dose of 0.02–0.05 mg twice a day is tolerated for 3–7 days, that dose is administered daily for 3–7 days and then discontinued.	Yes	0.6–1.2 mg
Ohio Children`s Hospital Neonatal Research Consortium 2013 [25]	0.05 q6 h	↑ to 0.1 q6 h (score > 8 after 3 doses) -> to 0.15 q6 h (score >8 after 3 doses)	Dose that keeps scores <8 for minimum of 48 h.	Decrease dose by 10% q24 h and discontinue when dose is <0.02	Yes	0.2
**Dosing Regimens and Current Practice From Literature**						
van Donge et al., 2019 [10]	Day 1: 0.1 q6 h	-	-	Day 2: 0.1 q12 h, Day 3: 0.05 q12 h, Day 4–7: 0.01 q24 h	No	0.4
McQueen et al., 2016 [26]	0.05 (score > 8 on 2 occasions or 1 score of ≥12)	↑ by 0.02 (score ≥12)	Dose that keeps scores < 8 for minimum 48 h	Decrease dose by 10% q24 h and discontinue 72 h after withdrawal	Yes	0.05
Brown et al., 2015 [27]	0.05 q4 h (score ≤ 12) 0.1 q4 h (score > 12)	-	0.05 q12 h with maximum of 0.2	Decrease dose by 10% if score ≤8 for q24 h	Yes	0.3–0.6
Lai et al., 2017 [28]	0.1 q6 h (score > 8)	-	When score < 8 for 1–2 days, 0.1 q12 h	0.1 q24 h	Yes	0.4
Napolitano et al., 2013 [29]	0.1 q12 h	↑ by 0.05 mg/kg q48 h with maximum of 1 mg/kg/day	-	Decrease dose by 10% at 1–2 week intervals	Yes	0.2
Raffaeli et al., 2017 [30]	0.05–0.1 q12 h	↑ dose by 10% q24 h-48 (score ≥ 12) maximum dose of 1 mg/kg/day	Dose that maintains score 9–11	Decrease dose by 10% q24 h if score ≤ 8	Yes	0.1–0.2
Siu et al., 2014 [31]	0.05–0.1 q6 h-q24 h	-	-	Decrease dose by 10–20% every week	No	0.05–0.4
Davis et al., 2018 [32]	Score 8–10 0.3 q8 h Score 11–13 0.5 q8 h Score 14–16 0.7 q8 h Score ≥ 17 0.8 q8 h	-	-	Decrease maintenance dose by 10% q12 h-q48 h and discontinue when dose is 20% of initial dose	Yes	0.9–2.4
Wiles et al., 2015 [18]	8 taper steps: 0.05 q6 h, 0.04 q6 h, 0.03 q6 h, 0.02 q6 h, 0.02 q8 h, 0.02 q12 h, 0.01 q12 h, 0.0.1 q24 h	If infant fails step 1 (score ≥ 12) consider 0.1 q6 h, 0.075 q6 h and 0.05 q6 h.	If average score is 8–12 do not wean.	Wean to the next step if average score is <8 for the past 24 h and discontinue after observations for 72 h from the last dose of step 8.	Yes	0.2
Hall et al., 2015 [33]	8 taper steps: 0.1 q6 h, 0.07 q12 h, 0.05 q12 h, 0.04 q12 h, mg/kg q12 h, 0.02 q12 h, 0.01 q12 h, 0.01 q24 h	If infant fails step 1 (score ≥ 12) consider 0.1 mg/kg q4 h, 0.1 mg/kg q8 h, 0.1 mg/kg q12 h.	If average score is 8–12 do not wean.	Wean to the next step if average score is <8 for past 24 h and discontinue after observations for 72 h from the last dose of step 8.	Yes	0.4

* Doses are mg/kg unless otherwise specified. ^$^ Total daily dose in the initial phase. NAS Neonatal abstinence syndrome↑increase.

**Table 2 children-08-00174-t002:** Overview of predicted cumulative area under the concentration-time curve (AUC) and predicted plasma concentration (assessed as Cmin) during the first 72 h after birth for three different gestational age groups. Applying van Donge et al. dosing regimen [10].

		12 h after Birth	24 h after Birth	48 h after Birth	72 h after Birth
Gestational age 28 weeks					
Start at 12 h	AUC	399 [285–538]	883 [633–1187]	2167 [1566–2900]	3557 [2609–4718]
	Cmin	31.1 [23.2–42.7]	42.6 [30.8–56.9]	54.3 [40.0–71.9]	54.9 [40.9–71.7]
Start at 36 h	AUC	399 [285–538]	766 [553–1047]	1539 [1137–2067]	2729 [2032–3673]
	Cmin	31.1 [23.2–42.7]	29.6 [22.0–39.7]	38.3 [28.7–51.5]	50.9 [38.6–68.1]
Gestational age 32 weeks					
Start at 12 h	AUC	380 [282–505]	838 [621–1112]	2037 [1520–2688]	3269 [2442–4253]
	Cmin	29.5 [21.9–38.8]	39.8 [29.9–52.5]	49.0 [36.8–63.4]	46.0 [34.7–59.7]
Start at 36 h	AUC	380 [282–505]	716 [548–948]	1393 [1074–1836]	2498 [1913–3244]
	Cmin	29.5 [21.9–38.8]	25.9 [19.7–33.7]	34.4 [26.5–44.7]	45.0 [34.7–58.8]
Gestational age 36 weeks					
Start at 12 h	AUC	361 [274–478]	811 [630–1054]	1993 [1559–2585]	3141 [2472–4077]
	Cmin	27.5 [21.0–35.7]	38.8 [30.3–50.1]	45.9 [36.0–60.0]	40.7 [31.2–52.7]
Start at 36 h	AUC	361 [274–478]	671 [505–864]	1283 [961–1654]	2361 [1818–3017]
	Cmin	27.5 [21.0–35.7]	22.5 [16.9–29.8]	32.7 [25.3–41.6]	43.0 [33.0–54.4]

AUC area under the concentration–time curve (ng·h/mL). Cmin minimum concentration (ng/mL). Data presented as median [interquartile range, IQR].

**Table 3 children-08-00174-t003:** Overview of predicted cumulative AUC during the first 72 h after birth for three different gestational age groups. Applying Neofax minimum and maximum dosing regimens [21].

Cumulative AUC (ng·h/mL)	12 h after Birth	24 h after Birth	48 h after Birth	72 h after Birth
Gestational age 28 weeks				
Neofax min	399 [285–538]	809 [583–1083]	1563 [1155–2080]	2281 [1694–3030]
Neofax max	399 [285–538]	920 [659–1241]	2312 [1663–3107]	4092 [2996–5471]
Gestational age 32 weeks				
Neofax min	380 [282–505]	756 [561–991]	1401 [1061–1833]	1965 [1501–2601]
Neofax max	380 [282–505]	880 [657–1168]	2188 [1631–2886]	3826 [2873–5028]
Gestational age 36 weeks				
Neofax min	361 [274–478]	704 [526–926]	1279 [958–1675]	1755 [1322–2299]
Neofax max	361 [274–478]	863 [654–1132]	2194 [1666–2845]	3786 [2895–4928]

AUC area under the concentration–time curve (ng·h/mL). Data presented as median [IQR].

## Data Availability

The data presented in this study are available on request from the corresponding author.

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
