# Peer review of "Pharmacometric Evaluation of Umbilical Cord Blood Concentration-Based Early Initiation of Treatment in Methadone-Exposed Preterm Neonates"

_children, 2021, doi:10.3390/children8030174_

Round 1
Reviewer 1 Report
Review for Children
Enhanced Care of Methadone-Exposed Preterm Neonates with Early Identification of Patients at Risk and Use of a Pharmacokinetic-Guided Dosing Regimen.
- Overall, the topic of the paper is novel with limited research performed on this particular topic. Unfortunately, there are many assumptions made in order to complete the study and some of reasoning behind the overall concept is not supported by current standards.
Introduction
- “Although methadone is considered the first-line drug for the treatment of NAS in many countries9 ,” – this statement is not true in the US and the author keeps jumping from recommendations in the world, US, etc. The standards of treatment are different in different countries and you would need to use the AAP recommendations if you are including US in the list of “many countries”.
- Reference provided, #9, is for Canada and does not represent “many countries”, also it states that methadone and morphine are medications of choice and not methadone.
- Initially spoke about NAS, which is really only defined for term neonates and in paragraph 3 switched to speaking about preterm. Transition difficult to follow or understand.
- Author may want to look at this reference to add some information regarding preterm withdrawal: Gibson KS, Stark S, Kumar D, Bailit JL. The relationship between gestational age and the severity of neonatal abstinence syndrome. Addiction. 2017;112(4):711-716.
- Based on Ruwanpathirana R, Abdel-Latif ME, Burns L, et al. Prematurity reduces the severity and need for treatment of neonatal abstinence syndrome. Acta Paediatr. 2015;104(5):e188-194 preterm withdrawal is diminished and it is difficult to compare it to withdrawal in term babies.
- “For the purpose of the current simulation study, we defined preterm neonates at risk for severe NAS as neonates £36 weeks GA with assumed umbilical cord methadone concentrations of < 60 ng/mL15”. Why did you choose 36 weeks, if 37 weeks is considered term? It is known that late preterm and term do not behave in the same manner, therefore why not 35 weeks to include all late preterm?
- Also, why 60? If you state, “. Furthermore, none of the infants with postnatal plasma methadone concentrations ≥20 ng/mL at 48 hours of age required pharmacologic treatment for their withdrawal symptoms15. “, why not use 20? Why 60?
Materials and Methods
- “We applied the population PK model of van Donge et al. as the population investigated in this study was exclusively preterm neonates (n=29) and methadone was administered orally, shortly after birth (median postnatal age= 3 days)10”.
- How did you come up with 29? How were these patients selected? Approached? Methods for this portion not presented.
- “demographic characteristics of three typical preterm neonates (28, 32, 36 weeks of GA and 1.25, 1.6, 2.3 kg birth weight, respectively) were used”
- how did you pick these gestational ages and weights? Poorly explained.
- “Treatment initiation time was set to 12 hours and 36 hours after birth and treatment duration to 72 hours”
- How did you decide to “set time to 12 and 36 hours”? Many of the babies exposed to opiates withdraw later then 12 and even 36 hours, depending on the drug. Typical withdrawal from methadone begins at approximately 48-72 hours (Prabhakar Kocherlakota, Pediatrics, 8/2014).
- Why is treatment duration up to 72 hours? That time period does not make sense to me and there is inadequate explanation.
Discussion
- “Methadone-associated NAS often requires prolonged hospitalization and high postnatal cumulative opioid treatment. Evidence has shown that delayed initiation of treatment and lack of optimal dosing regimen are two important factors influencing the care of NAS disease 8,34,35” -
- not really what those references are stating. Lack of optimal dosing is a definite factor, but not really an explanation for prolonged hospitalization. This is a bit of a just and would benefit from further explanation.
- “Limited data in opioid-exposed term neonates, have suggested the correlation of umbilical cord methadone concentration with incidence and severity of NAS in need for pharmacologic treatment14-16”
- This is data on term neonates and it is limited at best. How did you decide that you can use it for preterm and especially preterm of different gestational ages, including ELBW?
- “Such early initiation of treatment requires a validated tool for an early identification of patients at high risk for NAS disease in need for pharmacologic treatment. The utility of umbilical cord methadone concentration, in development of such tool, is a highly interesting and important area in need for prospective investigations in both preterm and term methadone-exposed neonates.”
- Can compare your tool with Finnegan for term, but not for preterm, since Finnegan should not be used in preterm at all.
- Currently, the push in the world is to use less medication for withdrawal, how would you assure that you will not overtreat these babies and provide medication for babies who have some withdrawal, but would not have been started on meds if monitored according to current standards.
- “Furthermore, in view of lack of data specific to preterm neonates, extrapolation of data is unavoidable to expand the evidence required to optimize the care of such vulnerable population38. “
- This is an assumption and since clearance is very different between a 25 week preterm baby and a 35 week baby, nothing was said to address these differences.
Author Response
Prof. Dr. Sari A. Acra,
Editor-in-Chief
Division of Gastroenterology, Hepatology, and Nutrition, Vanderbilt Children’s Hospital, 2200 Children’s Way, 9214 Doctors’ Office Tower, Nashville, TN 37232-9175, USA
Dear Dr. Acra,
I am writing this letter in regard to our revised manuscript entitled, “Enhanced Care of Methadone-Exposed Preterm Neonates with Early Identification of Patients at Risk and Use of a Pharmacokinetic-Guided Dosing Regimen.” , Manuscript ID: children-1115359, for consideration of publication as a Children research article.
We are grateful for the valuable feedback from the reviewers, which has helped us to improve the quality of our manuscript. Below we have carefully responded to all comments.
Sincerely,
Samira Samiee-Zafarghandy, MD, FRCPC
samiees@mcmaster.ca
Reviewer 1
Enhanced Care of Methadone-Exposed Preterm Neonates with Early Identification of Patients at Risk and Use of a Pharmacokinetic-Guided Dosing Regimen.
Overall, the topic of the paper is novel with limited research performed on this particular topic. Unfortunately, there are many assumptions made in order to complete the study and some of reasoning behind the overall concept is not supported by current standards.
Introduction
- “Although methadone is considered the first-line drug for the treatment of NAS in many countries9 ,” – this statement is not true in the US and the author keeps jumping from recommendations in the world, US, etc. The standards of treatment are different in different countries and you would need to use the AAP recommendations if you are including US in the list of “many countries”.
Based on this comment, we have made the following revision and have also changed the reference from Canadian Pediatric Society Statement to the American Academy of Pediatrics Statement:
Page 2, Line 15
- Although methadone is considered the first-line drug for the treatment of NAS in many countries,
Changed to:
- Although methadone is one of the most common opioids, used as first-line treatment for NAS,
- Reference provided, #9, is for Canada and does not represent “many countries”, also it states that methadone and morphine are medications of choice and not methadone.
Please see our response to comment number 1.
- Initially spoke about NAS, which is really only defined for term neonates and in paragraph 3 switched to speaking about preterm. Transition difficult to follow or understand.
We have revised our introduction, so that it focuses on preterm neonates and provides a comprehensible background information and well-ordered transition.
- Author may want to look at this reference to add some information regarding preterm withdrawal: Gibson KS, Stark S, Kumar D, Bailit JL. The relationship between gestational age and the severity of neonatal abstinence syndrome. Addiction. 2017;112(4):711-716.
Thank you for this suggestion. We have now incorporated this article’s data to our manuscript to enhance our introduction in regard to the insufficiency of the Finnegan Neonatal Abstinence Scoring (FNAS) Tool for assessment of NAS severity in preterm neonates and the urgent need for an alternative approach.
- Based on Ruwanpathirana R, Abdel-Latif ME, Burns L, et al. Prematurity reduces the severity and need for treatment of neonatal abstinence syndrome. Acta Paediatr. 2015;104(5):e188-194 preterm withdrawal is diminished and it is difficult to compare it to withdrawal in term babies.
We agree with the reviewer that physiologic immaturity can influence how and to what degree symptoms of NAS manifest in preterm neonates. However, as this article suggests, it is possible that the perceived diminished withdrawal symptoms in preterm neonates, result from lack of accurate assessment and under-diagnosis. The available evidence suggests the inadequacy of the FNAS tool for accurate assessment of withdrawal symptoms in preterm neonates and therefore, it is of utmost importance to explore adjunctive measures that can provide accurate assessment or additional information on the risk for development of NAS, such as electroencephalography to assess sleep activity, movement assessment with actigraphy, and salivary cortisol level as suggested by Ruwanpathirana R et al., or cord blood methadone concentration as suggested by our current manuscript.
- “For the purpose of the current simulation study, we defined preterm neonates at risk for severe NAS as neonates £36 weeks GA with assumed umbilical cord methadone concentrations of < 60 ng/mL15”. Why did you choose 36 weeks, if 37 weeks is considered term? It is known that late preterm and term do not behave in the same manner, therefore why not 35 weeks to include all late preterm?
We used neonates of ≤ 36+6 weeks to include all preterm neonates, including the late preterm ones. Furthermore, as we applied the population PK model of van Donge et al, we chose a population for the current study (preterm population <= 36+6) that reflects best the population included in the initial study by van Donge et al. which had a mean gestational age of 32 weeks (range 26 to 36).
- Also, why 60? If you state, “. Furthermore, none of the infants with postnatal plasma methadone concentrations ≥20 ng/mL at 48 hours of age required pharmacologic treatment for their withdrawal symptoms15. “, why not use 20? Why 60?
As per the available evidence, the above concentrations, at delivery (60 ng/mL) and 48 hours of life (20 ng/mL), reflect postnatal clearance after interruption of exposure:
- Methadone-exposed neonates with plasma methadone concentration of ≥ 60 mg/mL on day zero of life did appear to be protected from NAS (Rosen TS, Pippenger CE. doi:10.1016/s0022-3476(76)81074-8)
- All infants who required treatment had cord concentrations < 53 ng/ml (Kuschel CA, et al. doi:10.1136/adc.2003.036863).
- None of the infants with postnatal plasma methadone concentrations ≥20 ng/mL at 48 hours of age required pharmacologic treatment for their withdrawal symptoms (Kuschel CA, et al. doi:10.1136/adc.2003.036863).
In our simulation model, we focused on the umbilical cord methadone concentration as it provides the “first” methadone concentration, in neonates, in a “non-invasive” approach. Both of the above factors are important for development of a potential adjunctive tool that can provide feasible and “timely” information, in methadone-exposed premature neonates, assessing the risk for development of NAS in need for pharmacologic treatment.
Materials and Methods
- “We applied the population PK model of van Donge et al. as the population investigated in this study was exclusively preterm neonates (n=29) and methadone was administered orally, shortly after birth (median postnatal age= 3 days)10”.
- How did you come up with 29? How were these patients selected? Approached? Methods for this portion not presented.
We thank the reviewer for highlighting this point. We have now provided additional details on the referred study and its population pharmacokinetic model (van Donge et al., DOI: 10.1111/bcp.13906) that has been applied to this current manuscript.
For further clarity, we have also made the following change to our manuscript
Page 2, Line 14
- We applied the population PK model of van Donge et al. as the population investigated in this study was exclusively preterm neonates (n=29) and methadone was administered orally, shortly after birth (median postnatal age= 3 days)10. The median body weight and GA of the population of this study was 1.6 kg and 32 weeks, respectively. The data were best described by a one-compartment model with linear elimination. Clearance and volume of distribution were both influenced by GA. Apparent volume of distribution and clearance for (R)-methadone and (S)-methadone was estimated to be 26.9 L and 0.244 L/h, and 18 L and 0.167 L/h, respectively.
Changed to:
- We applied the population PK model of van Donge et al. as the population investigated in this study was exclusively preterm neonates and methadone was administered shortly after birth (median postnatal age of 3 days)10. PK data of this single-centre open-label prospective study included 29 preterm neonates which were exposed to a single dose of orally administered methadone. One dose of standard opioid medication (fentanyl or morphine) which was prescribed for clinical reasons was replaced by one dose of 0.1 mg/kg methadone. The median gestational age and body weight amounted 32 weeks (range 26 to 36) and 1.6 kg (range 0.93-2.7kg), respectively. The data were best described by a one-compartment model with linear elimination. Clearance and volume of distribution were both influenced by GA. Apparent volume of distribution and clearance for (R)-methadone and (S)-methadone was estimated to be 26.9 L and 0.244 L/h, and 18 L and 0.167 L/h, respectively. For further details on the population PK model, we refer to the initial study10.
- “demographic characteristics of three typical preterm neonates (28, 32, 36 weeks of GA and 1.25, 1.6, 2.3 kg birth weight, respectively) were used”
- how did you pick these gestational ages and weights? Poorly explained.
Additional information has been added to this part of the Method section. We selected the above three gestational age (GA) groups (28, 32 and 36 gestational age), their relevant birth weight (BW), and pharmacokinetic data from the initial study (median (range) GA of 32 (26-36) weeks and BW of 1.6 (0.93-2.7) kg). As risk of preterm birth (< 37 week) and very preterm birth (<32 week) is significantly higher in methadone-exposed pregnancies, these three GA groups, probably provides data on the majority of premature infants at risk for development of NAS (Cleary et al., DOI: 10.1016/j.ajog.2010.10.004).
The relationship between GA, body weight and PNA are highly correlated and we have selected the median patient from the initial study as our typical patient in the current study characterized by the specific GA and correlated BW, in order to reflect the patient population as closely as possible.
To further clarify this, we have made the following change to our manuscript
Page 3, Line 28
- “To generate individual methadone concentration-time profiles, the demographic characteristics of three typical preterm neonates (28, 32, 36 weeks of GA and 1.25, 1.6, 2.3 kg birth weight, respectively) were used.”
Changed to:
- “To generate individual methadone concentration-time profiles, the demographic characteristics of three typical preterm neonates (28, 32, 36 weeks of GA and 1.25, 1.6, 2.3 kg birth weight, respectively) were used. The demographic characteristics of GA and body weight of the patients in the original study (median of 32 weeks and 1.6 kg) were selected as we aimed to use their corresponding PK characteristics in our current model.”
- “Treatment initiation time was set to 12 hours and 36 hours after birth and treatment duration to 72 hours”
- How did you decide to “set time to 12 and 36 hours”? Many of the babies exposed to opiates withdraw later then 12 and even 36 hours, depending on the drug. Typical withdrawal from methadone begins at approximately 48-72 hours (Prabhakar Kocherlakota, Pediatrics, 8/2014).
In this study, one of our aims was to explore how the PK parameters of the commonly used dosing regimens relate to the available exposure-response data. Therefore, we chose to investigate two hypothetical treatment initiation times and assessed how quickly the particular target concentration was reached. The choice of these two initiation times (12 and 36 hours post birth) was as per the below evidence:
It is suggested that early identification and aggressive treatment of NAS may help to decrease severity and length of stay (Thigpen et al., doi: 10.5863/1551-6776-19.3.144). It is also shown that an association of severe central nervous system signs with the rate of decline of neonatal plasma methadone concentrations between postnatal days 1 and 4 (Doberczak et al., 10.1136/adc.2003.036863). Furthermore, in a study by Kuschel et al. (Arch Dis Child Fetal Neonatal Ed 2004;89:F390–F393) it was shown that pharmacologic treatment of NAS was instituted at a median age of 35 hours.
We therefore chose to simulate and compare an initiation of treatment at two time points:
- 12 hours after birth: The time that provides opportunity for early treatment initiation (with potential prevention of rapid decline of the neonatal plasma methadone concentration) but also access to cord blood methadone concentration and a proper assessment of the newborn for signs of central nervous system irritability can be available.
- 36 hours birth: The time that is reported in the literature as the median postnatal age for initiation of pharmacologic treatment.
To further clarify this point, we have made the following revision to our manuscript:
Page 2, Line 42:
- “Treatment initiation time was set to 12 hours and 36 hours after birth and treatment duration to 72 hours.”
Changed to:
- “Treatment initiation time was set to 12 hours and 36 hours after birth and treatment duration to 72 hours, to simulate early treatment initiation (12 hours) versus standard practice (36 hours). The early treatment initiation was chosen at 12 hours after birth to ensure feasibility of access to umbilical cord methadone concentration as a potential predictor. Simulations were performed using nonlinear mixed effects modelling software (NONMEM v7.4.1; ICON Development Solutions, Ellicott City, MD)26. We investigated the plasma concentration and AUC within the 72 hours after birth, as surrogate of response for each of the assessed dosing regimens at the two selected treatment initiation times.”
- Why is treatment duration up to 72 hours? That time period does not make sense to me and there is inadequate explanation.
The focus of the current study was on the first initial 72 hours of methadone treatment, for the following reasons:
- i) The available data on target exposure for AUC and plasma concentration from the limited available studies in term population, were all within the first 72 hours of life. These target exposures were identified as possible prognosticators for development of NAS in need for treatment.
- ii) The rate of decline in methadone concentration is suggested as a factor for development of NAS in need for pharmacologic treatment. In the current study, we aimed to simulate the typical initiation time (36 hours post birth) vs an early initiation time (12 hours post birth) of treatment to depict the change in methadone concentration within the first 72 hours of life, where symptoms were withdrawal symptoms typically start.
The focus of the current study was the timing of the initiation of the methadone treatment and whether umbilical cord could serve as a potential non-invasive approach to determine whether the patient would require treatment or not. In addition, we wanted to study the exposure of methadone when an in utero exposure was assumed. For these reasons, the main focus was not the investigation of the treatment duration and this was set to three days. Also, since the NAS scoring system was not included in the model (although pharmacodynamic response was assessed by AUC), no assessment or conclusion could be made on different treatment durations. Since many (almost all) dosing regimen depend on the NAS scoring system, which is inappropriate in preterm neonates and is highly subjective, we advocate for new clinical studies focussing on the PD assessment of NAS in preterm neonates, in order that exposure (PK) could be linked to the response (PD).
Discussion
- “Methadone-associated NAS often requires prolonged hospitalization and high postnatal cumulative opioid treatment. Evidence has shown that delayed initiation of treatment and lack of optimal dosing regimen are two important factors influencing the care of NAS disease 8,34,35” -
- not really what those references are stating. Lack of optimal dosing is a definite factor, but not really an explanation for prolonged hospitalization. This is a bit of a just and would benefit from further explanation.
It is suggested that early identification and aggressive treatment of NAS may help to decrease severity and length of stay (Thigpen et al., doi: 10.5863/1551-6776-19.3.144). Use of pharmacokinetic-modeled methadone dosing regimen as compared to standard methadone weaning has also shown to reduce duration of opioid exposure and length of stay (Eris S. Hall et al., doi.org/10.1016/j.jpeds.2015.09.038; van Donge et al., DOI: 10.1111/bcp.13906)
As per the reviewer’s comment, we revised this paragraph and the relevant referencing as below:
Page 7, Line 22:
- “Methadone-associated NAS often requires prolonged hospitalization and high postnatal cumulative opioid treatment. Evidence has shown that delayed initiation of treatment and lack of optimal dosing regimen are two important factors influencing the care of NAS disease.”
Changed to:
- “Methadone-associated NAS often requires prolonged hospitalization and high postnatal cumulative opioid treatment. Evidence has shown early identification of neonates in need for pharmacological treatment and optimal dosing strategy may decrease the severity of NAS and length of stay.”
- “Limited data in opioid-exposed term neonates, have suggested the correlation of umbilical cord methadone concentration with incidence and severity of NAS in need for pharmacologic treatment14-16”
- This is data on term neonates and it is limited at best. How did you decide that you can use it for preterm and especially preterm of different gestational ages, including ELBW?
We agree with the reviewer’s comment that the data on predictability of cord methadone concentration and postnatal methadone concentration for severity of NAS is limited and only available in term neonates.
In the current study, our overarching aim has been to generate data on pharmacokinetic (plasma concentration, area under the concentration-time curve)-pharmacodynamic (NAS scores, need for pharmacologic treatment) of methadone in preterm neonates. Although for this purpose, we had to extrapolate the available data in regard to PK-PD relationship of methadone in term neonates with NAS, we applied the pharmacokinetic characteristics of preterm neonates from a study by van Donge et al., in which the data was obtained from administration of oral methadone in 29 preterm neonates with gestational age of 26 to 36 wks. In the absence of PK-PD relationship evidence in preterm neonates, extrapolation of data from term neonates is justified. Specifically as limited available evidence on cord plasma concentrations of methadone-exposed term and preterm neonates did not show any significant difference (de Castro et al., 10.1097/FTD.0b013e31822724f0).
Furthermore, we have included this limitation in the discussion of our study and recommended a prospectively designed study to validate this PK-PD relationship in preterm neonates.
“Such early initiation of treatment requires a validated tool for an early identification of patients at high risk for NAS disease in need for pharmacologic treatment. The utility of umbilical cord methadone concentration, in development of such tool, is a highly interesting and important area in need for prospective investigations in both preterm and term methadone-exposed neonates.”
- Can compare your tool with Finnegan for term, but not for preterm, since Finnegan should not be used in preterm at all.
We agree with the Reviewer FNAS tool has not been validated in preterm neonates and ideally it should not be applied to this population. In the current proposal, we recommend umbilical cord methadone concentration as a feasible approach for development of a tool that can be applied to preterm neonates at risk for development of NAS. We recommend a prospectively designed study to validate our findings in the development of such tool and provide comparison with use of Finnegan score.
- Currently, the push in the world is to use less medication for withdrawal, how would you assure that you will not overtreat these babies and provide medication for babies who have some withdrawal, but would not have been started on meds if monitored according to current standards.
We thank the reviewer for this comment. We agree that sole use of umbilical cord methadone concentration could have a risk of overtreatment. Therefore, we have emphasized on evaluating the performance of this tool, per se or as an adjunctive tool, for risk prediction of NAS in a prospective study.
- “Furthermore, in view of lack of data specific to preterm neonates, extrapolation of data is unavoidable to expand the evidence required to optimize the care of such vulnerable population38. “
- This is an assumption and since clearance is very different between a 25 week preterm baby and a 35 week baby, nothing was said to address these differences.
We agree with the reviewer that we had to extrapolate the available data in regard to PK-PD relationship of methadone in term neonates with NAS, but we would like to emphasize that in the current study, we did use the available data on individual PK parameters of premature neonates and applied the PK characteristics of preterm neonates from a study by van Donge et al., in which the data was obtained from administration of oral methadone in 29 preterm neonates with gestational age of 26 to 36 wks. Therefore the above statement regarding the extrapolation, is related to the PD assessment and application of AUC target obtained from a term population. In the preterm PK model of van Donge et al., GA was identified as the main covariate impacting the clearance, but because the dosing is based on weight the influence of maturation is taken into account as well.

Reviewer 2 Report
This is an interesting and well conducted study that will add to the literature on an understudied cohort of patients with in utero opioid exposure. There is novelty in the approach and this brings the concept of exposure response to neonatal abstinence syndrome. I would suggest some more clarity around the arguments being made the assumptions used in support of the eventual conclusions. This clarity will ultimately provide the paper with more of an impact in the field.
The modeling assumes the in utero exposure is methadone. This should be explicitly stated, as well as any assumptions of other exposures an infant may have had in utero.
The approach used does not incorporate NAS scores (Finnegan or Eat, Sleep, Console) into the modeling. The authors suggest that not symptoms themselves, but infant plasma methadone of >60 ng/ml on day zero of life be used as a risk stratifier of “severe NAS” . Since one of the simulations has drug starting within 12 hours of birth, there is an implicit assumption that this would be a group of patients who would merit early pharmacologic intervention independent of symptoms scores, or based only on 3 sets of scores. Plasma methadone is a predictor, but there are other well defined risk factors for symptom severity (Patrick PMID 33080277). A stated goal of the paper is to prevent over-treatment (defined as pharmacologic therapy) of NAS. The treatment of the newborn within 12 hours before there has been emergence of NAS symptoms will end up using pharmacologic therapies on a number of infants who ultimately would not started on methadone using standard symptom based approaches.
An uncited assertion is made in the introduction is that the Finnegan scoring tool will result in delayed initiation of pharmacologic dosing. The citation 13 (Schiff and Grossman) used to support the assertion that this delay will result in high dose, prolonged opioid exposure and protracted length of stay. This is not quite correct, as Schiff actually argues that the Finnegan promotes the over use of pharmacologic therapies. There is not consensus in the literature that the Finnegan scoring system delays the identification of infants who would benefit from pharmacologic treatment.
Defining “severe NAS” in preterms is also a challenge given differences in expression of signs compared to term infants (sleep and Moro for example). This makes the link to a specific newborn day zero methadone concentration as proxy for worse NAS more tenuous. This should be noted.
The authors point out the variety of published methadone regimens, there are certainly many more local variations which have not been published. Young and Devlin ( Pediatrics. 2021;147(1):e2020008839) document the huge variability in the US of approaches and outcomes in NAS care. Some sites treat only 6% of infants with pharmacologic means while others treat 100%. The variation in trigger scores, second agent of choice and a host of other factors indicate there is far from consensus on these points. Large variation in medicine almost always identifies a lack of evidence. For NAS this lack of evidence is also indicated by lack of consensus of what amount of withdrawal symptoms clinicians should be willing to tolerate. The authors are entering the debate by tying treatment to a proxy for symptoms, when the amount of symptoms we as clinicians should tolerate before starting treatment has not been established.
The target concentration-time curves (AUCs) at 24 and 48 hours of 816 and 2274 ng·h/mL, respectively from Wiles were used as target, though this was derived from term infants. There should be ore of a discussion of the suitability of this goal to preterm infants. Rosen (ref 14) suggested amioloration of symptoms at methadone concentration >60 ng/ml. This appears similar to that suggested by Wiles (30-50 ng/ml).
There is an assertion that current regimens can lead to under or overdosing of methadone. The yardstick that defines optimized dosing appears to be based solely upon the Wiles target. Given the unsteadiness of this assumption, the following line in conclusion should be softened. “Methadone-exposed preterm neonates are at risk for under- or over-treatment with use of currently recommended methadone dosing regimen”.
The methods describe appropriate use of different weights for each different age cohort. Were there any other assumptions about differential clearance between these ages or other factors which could impact drug exposure?. PBPK modeling has suggested CYP2B6 andCYP3A4 activity, α1-acid glycoprotein, and microsomal protein per gram of liver all have an effect on the predicted clearance of R- and S-methadone in neonates and adults (McPahil and Vinks JCP 2020). This is important given the known large inter-patient variability of methadone PK in neonates.
The use of umbilical cord methadone concentration as a biomarker of disease severity is a key part of this paper. This biomarker would need to predict the actual scores. Simply just measuring the scores themselves is much easier to determine. This point should be made more explicit. The presumptive advantage would be that early treatment before emergence of symptoms would change the natural history of withdrawal symptoms. This is not a concept that has been well supported empirically. The receiver operator curves for this biomarker ideally would also need to be defined. This is beyond the scope of this paper, but should be mentioned. Modeling could predict how use of cord blood may both control symptoms quicker, but also how many infants would be treated who would have not met standard of care symptom based triggers.
The title could also be more clear on the argument in the paper. “Enhanced care” should be removed. A possible suggested title would be “Pharmacokinetic modeling of biomarker based early initiation of pharmacologic treatment in Methadone-Exposed Preterm Neonates”
Author Response
Prof. Dr. Sari A. Acra,
Editor-in-Chief
Division of Gastroenterology, Hepatology, and Nutrition, Vanderbilt Children’s Hospital, 2200 Children’s Way, 9214 Doctors’ Office Tower, Nashville, TN 37232-9175, USA
Dear Dr. Acra,
I am writing this letter in regard to our revised manuscript entitled, “Enhanced Care of Methadone-Exposed Preterm Neonates with Early Identification of Patients at Risk and Use of a Pharmacokinetic-Guided Dosing Regimen.” , Manuscript ID: children-1115359, for consideration of publication as a Children research article.
We are grateful for the valuable feedback from the reviewers, which has helped us to improve the quality of our manuscript. Below we have carefully responded to all comments.
Sincerely,
Samira Samiee-Zafarghandy, MD, FRCPC
samiees@mcmaster.ca
Reviewer 2.
This is an interesting and well conducted study that will add to the literature on an understudied cohort of patients with in utero opioid exposure. There is novelty in the approach and this brings the concept of exposure response to neonatal abstinence syndrome. I would suggest some more clarity around the arguments being made the assumptions used in support of the eventual conclusions. This clarity will ultimately provide the paper with more of an impact in the field.
- The modeling assumes the in utero exposure is methadone. This should be explicitly stated, as well as any assumptions of other exposures an infant may have had in utero.
We thank the reviewer for this comment. This information is added to the methods and highlighted again discussion.
Page 2, Line 32
“We assumed a neonatal methadone concentration approximating 30 ng/mL at birth to target the population that have the highest risk for the development of NAS and are in need for pharmacological intervention (<60 ng/mL).”
Changed to:
“We assumed an in utero exposure to methadone only with a neonatal methadone con-centration approximating 30 ng/mL at birth to target the population that have the highest risk for the development of NAS and are in need for pharmacological intervention (<60 ng/mL).”
- The approach used does not incorporate NAS scores (Finnegan or Eat, Sleep, Console) into the modeling. The authors suggest that not symptoms themselves, but infant plasma methadone of >60 ng/ml on day zero of life be used as a risk stratifier of “severe NAS” . Since one of the simulations has drug starting within 12 hours of birth, there is an implicit assumption that this would be a group of patients who would merit early pharmacologic intervention independent of symptoms scores, or based only on 3 sets of scores. Plasma methadone is a predictor, but there are other well defined risk factors for symptom severity (Patrick PMID 33080277). A stated goal of the paper is to prevent over-treatment (defined as pharmacologic therapy) of NAS. The treatment of the newborn within 12 hours before there has been emergence of NAS symptoms will end up using pharmacologic therapies on a number of infants who ultimately would not started on methadone using standard symptom based approaches.
We thank the reviewer for this comment. The Reviewer highlighted correctly that since no information on the NAS scores were available in the initial study we were not able to include the NAS score in the development of the current population PK model. We, however, extrapolated the available data on relationship of plasma concentration and AUC to NAS score severity from studies in term neonate, to our preterm physiologic based PK model (using PK data obtained from premature neonates of study by van Donge et al.). We believe as a first study, to generate data on utility of umbilical cord methadone concentration as a tool to predict neonates at risk of development of NAS in need for pharmacologic treatment, this extrapolation of data while maintain a model developed as per the preterm physiology is novel and informative.
We also agree with the reviewer that sole use of umbilical cord methadone concentration could have a risk of overtreatment. Therefore, we have emphasized on evaluating the performance of this tool for risk prediction of NAS in a prospective study per se and as an adjunctive tool with highest sensitivity, specificity, negative and positive predictive value. It is of importance to include the various scoring systems in this study as well. This concern is highlighted in the discussion.
Page 9, Line 27
- “Such early initiation of treatment requires a validated tool for an early identification of patients at high risk for NAS disease in need for pharmacologic treatment. The utility of umbilical cord methadone concentration, in development of such tool, is a highly interesting and important area in need for prospective investigations in both preterm and term methadone-exposed neonates.
Changed to:
- “Such early initiation of treatment requires a validated tool for an early identification of patients at high risk for NAS disease in need for pharmacologic treatment. The utility of umbilical cord methadone concentration, in development of such tool, is a highly interesting and important area in need for prospective investigations in both preterm and term methadone-exposed neonates. The inclusion of various scoring systems in these suggested studies are essential to relate the exposure of methadone to the clinical response.”
- An uncited assertion is made in the introduction is that the Finnegan scoring tool will result in delayed initiation of pharmacologic dosing. The citation 13 (Schiff and Grossman) used to support the assertion that this delay will result in high dose, prolonged opioid exposure and protracted length of stay. This is not quite correct, as Schiff actually argues that the Finnegan promotes the over use of pharmacologic therapies. There is not consensus in the literature that the Finnegan scoring system delays the identification of infants who would benefit from pharmacologic treatment.
We thank the reviewer for this comment. We have now addressed this comment by clarifying that use FNAS tool might delay the identification and initiation of treatment in preterm neonates, because of their physiologic immaturity.
Page 2, Line 27
- “It has been suggested that FNAS-driven interventions can result in delayed initiation of the pharmacological treatment.”
Changed to:
- “It has been suggested that FNAS-driven interventions can result in delayed initiation of the pharmacological treatment in preterm neonates, whose hunger-related cues maybe suppressed.”
- Defining “severe NAS” in preterm is also a challenge given differences in expression of signs compared to term infants (sleep and Moro for example). This makes the link to a specific newborn day zero methadone concentration as proxy for worse NAS more tenuous. This should be noted.
We agree with the reviewer that validation of any non-clinical tool for prediction of NAS in need for pharmacologic treatment will first need a valid clinical assessment that can be considered “Gold Standard” in preterm neonates. Currently the available withdrawal assessment tools, including the FNAS, is the strategy that universally being used for preterm and term neonates. We agree with the reviewer’s comment on significant limitation of this practice and have added the following statement to our discussion:
Page 9, Line 27
- Such early initiation of treatment requires a validated tool for an early identification of patients at high risk for NAS disease in need for pharmacologic treatment. The utility of umbilical cord methadone concentration, in development of such tool, is a highly interesting and important area in need for prospective investigations in both preterm and term methadone-exposed neonates.
Changed to
- “Such early initiation of treatment requires a validated tool for an early identification of patients at high risk for NAS disease in need for pharmacologic treatment. The utility of umbilical cord methadone concentration, in development of such tool, is a highly interesting and important area in need for prospective investigations in both preterm and term methadone-exposed neonates. The inclusion of various scoring systems in these suggested studies are essential to relate the exposure of methadone to the clinical response. Such clinical scoring system should be developed and validated, considering symptoms that are specific to preterm neonates for opioid withdrawal, while addressing other preterm medical diagnoses.”
- The authors point out the variety of published methadone regimens, there are certainly many more local variations which have not been published. Young and Devlin ( Pediatrics. 2021;147(1):e2020008839) document the huge variability in the US of approaches and outcomes in NAS care. Some sites treat only 6% of infants with pharmacologic means while others treat 100%. The variation in trigger scores, second agent of choice and a host of other factors indicate there is far from consensus on these points. Large variation in medicine almost always identifies a lack of evidence. For NAS this lack of evidence is also indicated by lack of consensus of what amount of withdrawal symptoms clinicians should be willing to tolerate. The authors are entering the debate by tying treatment to a proxy for symptoms, when the amount of symptoms we as clinicians should tolerate before starting treatment has not been established.
We agree with the reviewer that approach to NAS from many perspective (assessment, diagnosis or treatment) is variable due to lack of strong evidence. We believe our current study, can be a tool to assist optimizing individual practices. As the first step in the suggested prospective studies, the predictability of different umbilical cord methadone concentration for different clinical outcomes (No NAS, NAS managed by non-pharmacologic treatment, NAS managed by pharmacologic treatment) in methadone-exposed preterm neonates should be elucidated.
- The target concentration-time curves (AUCs) at 24 and 48 hours of 816 and 2274 ng·h/mL, respectively from Wiles were used as target, though this was derived from term infants. There should be ore of a discussion of the suitability of this goal to preterm infants. Rosen (ref 14) suggested amioloration of symptoms at methadone concentration >60 ng/ml. This appears similar to that suggested by Wiles (30-50 ng/ml).
We have now added further clarification in the discussion on use of target exposures from the available data in term neonates.
Page 8, Line 7
- “We observed the dosing recommendation of van Donge et al., which was the only dosing regimen designed with integration of developmental PK data in preterm neonates, provided cumulative exposures comparable to the suggested target exposures for optimal PK/PD relationship.”
Changed to:
- “We observed the dosing recommendation of van Donge et al., which was the only dosing regimen designed with integration of developmental PK data in preterm neonates, provided cumulative exposures comparable to the suggested target exposures, in term neonate, for optimal PK/PD relationship.”
Page 8, Line 42
- “The use of data on umbilical cord methadone concentrations, postnatal plasma concentrations and target exposures from available evidence in term methadone-exposed neonates can be considered a limitation.”
Changed to
- “The use of data on umbilical cord methadone concentrations, postnatal plasma concentrations and target exposures from limited available evidence in term methadone-exposed neonates is a limitation.”
- There is an assertion that current regimens can lead to under or overdosing of methadone. The yardstick that defines optimized dosing appears to be based solely upon the Wiles target. Given the unsteadiness of this assumption, the following line in conclusion should be softened. “Methadone-exposed preterm neonates are at risk for under- or over-treatment with use of currently recommended methadone dosing regimen”.
Thank you for this comment. We have made the following change to our conclusion:
Page 9, Line 9
- Methadone-exposed preterm neonates are at risk for under- or over-treatment with use of currently recommended methadone dosing regimen.
Changed to:
- Methadone-exposed preterm neonates can be at risk for under- or over-treatment with use of currently recommended methadone dosing regimen.
- The methods describe appropriate use of different weights for each different age cohort. Were there any other assumptions about differential clearance between these ages or other factors which could impact drug exposure?. PBPK modeling has suggested CYP2B6 andCYP3A4 activity, α1-acid glycoprotein, and microsomal protein per gram of liver all have an effect on the predicted clearance of R- and S-methadone in neonates and adults (McPahil and Vinks JCP 2020). This is important given the known large inter-patient variability of methadone PK in neonates.
We thank the Reviewer for highlighting this issue. Based on the applied population PK model from the study by van Donge et al., only gestational age was found to be an impacting factor on the clearance of methadone. In that particular study no information was collected on the genetic variants or polymorphisms of the patients. It is indeed known that polymorphisms of the CYP2B6 enzyme alter the disposition of methadone and influence the plasma concentrations, but this could not be addressed in this study since the data was not collected. The observed high interindividual variability in PK of methadone, not only in preterm neonates but also in adults, does not allow to easily disentangle the effect of genetic polymorphisms or other factors contributing to the high degree of interindividual variability.
- The use of umbilical cord methadone concentration as a biomarker of disease severity is a key part of this paper. This biomarker would need to predict the actual scores. Simply just measuring the scores themselves is much easier to determine. This point should be made more explicit. The presumptive advantage would be that early treatment before emergence of symptoms would change the natural history of withdrawal symptoms. This is not a concept that has been well supported empirically. The receiver operator curves for this biomarker ideally would also need to be defined. This is beyond the scope of this paper, but should be mentioned. Modeling could predict how use of cord blood may both control symptoms quicker, but also how many infants would be treated who would have not met standard of care symptom based triggers.
We also agree with the reviewer that sole use of umbilical cord methadone concentration could have a risk of overtreatment. Therefore, we have emphasized on evaluating the performance of this tool for risk prediction of NAS in a prospective study per se and as an adjunctive tool with highest sensitivity, specificity, negative and positive predictive value. It is of importance to include the various scoring systems in this study as well. This concern is highlighted in the discussion.
Page 8, Line 22:
- “Such early initiation of treatment requires a validated tool for an early identification of patients at high risk for NAS disease in need for pharmacologic treatment. The utility of umbilical cord methadone concentration, in development of such tool, is a highly interesting and important area in need for prospective investigations in both preterm and term methadone-exposed neonates.”
Changed to:
- “Such early initiation of treatment requires a validated tool for an early identification of patients at high risk for NAS disease in need for pharmacologic treatment. The utility of umbilical cord methadone concentration, in development of such tool, is a highly interesting and important area in need for prospective investigations in both preterm and term methadone-exposed neonates. The inclusion of various scoring systems in these suggested studies are essential to relate the exposure of methadone to the clinical response. Such clinical scoring system should be developed and validated, considering symptoms that are specific to preterm neonates for opioid withdrawal, while addressing other preterm medical diagnoses.” Evaluating the performance of this tool, per se and as an adjunctive tool, for risk prediction of NAS with highest sensitivity, specificity, negative and positive predictive value in a prospective study is of utmost importance.”
- The title could also be more clear on the argument in the paper. “Enhanced care” should be removed. A possible suggested title would be “Pharmacokinetic modeling of biomarker based early initiation of pharmacologic treatment in Methadone-Exposed Preterm Neonates”
- Enhanced Care of Methadone-Exposed Preterm Neonates with Early Identification of Patients at Risk and Use of a Pharmacokinetic-Guided Dosing Regimen.
Changed to:
- “Pharmacometric evaluation of an umbilical cord blood concentration based early initiation of treatment in methadone-exposed preterm neonates”.

Round 2
Reviewer 1 Report
Thank you for addressing all my concerns and rewording some of the more confusing sections. Even though there is not much literature on this topic, and the original assumption is mainly made based on 1 paper, overall, this is a very interesting look at this complex topic and relevant to the overall subject.